# Personalizing Child Protection: The Clinical Value and Usability of a Needs Assessment Instrument in The Netherlands

**DOI:** 10.3390/children9111702

**Published:** 2022-11-06

**Authors:** Anne M. E. Bijlsma, Mark Assink, Claudia E. Van der Put

**Affiliations:** Research Institute of Child Development and Education, University of Amsterdam, 1018 WS Amsterdam, The Netherlands

**Keywords:** child maltreatment, needs assessment, child protection services

## Abstract

Studies on child maltreatment prevention programs show that the effects of these programs are rather small. Drawing on the need principle of the Risk–Need–Responsivity model, program effects may be enhanced by properly assessing all the needs of individual families involved in child protection so that programs can be adapted to those needs. Recently, a needs assessment tool (ARIJ-Needs) has been developed in the Netherlands to support child protection practitioners in not only the assessment of treatment needs in individual families, but also in selecting the program(s) and/or intervention(s) that best target those needs. This study assessed the clinical value and usability of ARIJ-Needs by interviewing Dutch child protection practitioners (N = 15). A vignette describing a child protection case was used to examine differences between needs assessments based on unstructured clinical judgment (i.e., without using the assessment tool), and structured clinical judgment in which the assessment tool was used. The results showed that significantly more treatment needs were identified when ARIJ-Needs was used relative to clinical judgment in which ARIJ-Needs was not used. Specifically, needs related to parenting, the parent(s), and the family were identified more often when the assessment tool was used. This is an important finding, as these needs comprise the (changeable) risk factors that are most predictive of child maltreatment and should be addressed with priority to prevent child maltreatment. This study shows that ARIJ-Needs supports practitioners in assessing relevant needs in families at risk for child maltreatment. Study implications and recommendations for improvement of the ARIJ-Needs are discussed.

## 1. Introduction

Child maltreatment is a worldwide public health problem with serious consequences for the development of millions of children [1,2,3]. Therefore, effective interventions for reducing the risk for child maltreatment are highly needed. However, meta-analytic studies on the effectiveness of currently available treatment programs aimed at reducing child maltreatment show only small effects of these programs [4,5]. An explanation for these results may be that interventions are insufficiently personalized to the individual needs of families at risk [6,7]. A promising approach to improve personalized treatment in child protection services (CPS) is by applying the risk, need, and responsivity principles derived from the Risk–Need–Responsivity (RNR) model that was originally designed for the criminal justice system [8,9,10,11,12,13]. To facilitate the implementation of these principles in child protection, validated assessment instruments are highly needed [5]. Therefore, a child risk assessment instrument has recently been developed and validated in the Netherlands [12,14]. In addition, a needs assessment instrument was developed (ARIJ-Needs) to facilitate practitioners in adhering to the need principle in CPS [15]. This instrument was designed to support child protection practitioners in identifying personal needs of clients, and in selecting appropriate interventions that target those needs. To date, the clinical utility of this instrument has not been examined yet. Therefore, the main aim of this study was to examine whether ARIJ-Needs effectively supports child protection practitioners in their decisions on appropriate treatment programs based on their identification of care needs of at-risk families. In addition, insights into the current decision-making processes (without using the needs instrument) were gained to better determine the clinical value of the instrument.

The RNR model is a widely acknowledged and influential model in the criminal justice system that serves as a guidance for the assessment and treatment of offenders [8]. The model comprises three main principles: (1) the Risk principle stating that the intensity of treatment must be matched to an offender’s risk of criminal recidivism; (2) the Need principle stating that an offender’s criminogenic needs (i.e., the dynamic or changeable risk factors for criminal recidivism that are present) should be targeted; and (3) the Responsivity principle stating that interventions must be matched to an offender’s individual characteristics, such as learning style and abilities. The RNR model was originally developed for effectively reducing criminal involvement, but applying the RNR principles in child protection services may be very promising [12,13]. After all, criminal behavior and child maltreatment can both be explained by an interaction between risk factors (e.g., mental health problems) and protective factors (e.g., having strong social corrections) in various ecological systems such as the family, school, and neighborhood c.f. [15,16]. Furthermore, both delinquent behavior and occurrences of child abuse are determined by the balance between risk and protective factors [15,17,18,19,20,21,22].

In recent years, the first structured instruments became available that facilitate the implementation of the risk and need principles in child protection [13,15,23]. The ARIJ Safety assessment and ARIJ Risk assessment instruments have already been widely implemented in the Netherlands [14]. These instruments facilitate child welfare workers in the assessment of immediate safety of children and the risk for (the recurrence of) child maltreatment [12,14]. If there is a substantial risk of child maltreatment, a further assessment of dynamic (changeable) risk factors and personal needs of families is needed to provide insights into potential treatment targets. Recently, ARIJ-Needs has been developed to assess these treatment targets that comprise dynamic risk factors that are empirically associated with child maltreatment [15]. 

ARIJ-Needs consists of two components: (1) a needs-assessment component to assess risk factors that are dynamic in nature and can thus be changed in treatment (i.e., “needs”), and (2) a decision-making component to match the assessed needs to interventions that target those needs. The list of needs that are assessed in the needs-assessment component was based on a selection of significant predictors for child maltreatment derived from (meta-analytic) studies such as [21,24,25,26]. The selected dynamic risk factors were categorized into “parenting factors” (e.g., problematic parent–child interaction or inadequate supervision/monitoring), “family factors” (e.g., lack of social support or financial problems), “parent factors” (e.g., parental stress or criminal behavior), and “child factors” (e.g., internalizing problems or social problems). The decision-making component of the tool comprises a database of 116 interventions for (prevention of) child maltreatment that are available in the Netherlands and that is used in matching the needs to the interventions. These interventions target at least one of the need factors that are assessed in the needs-assessment component of the instrument [15].

It was expected that ARIJ-Needs enhances a more effective, efficient, and less subjective decision-making process of child protection practitioners by providing a structured needs assessment instrument that also provides an overview of the interventions that target the assessed needs. The aims of this study were to (1) gain insights into current practitioner decision-making processes of child protection practitioners to determine the added value of using ARIJ-Needs in assessing needs and selecting appropriate interventions, and (2) examine the usability of ARIJ-Needs in supporting practitioners. 

## 2. Methods

### 2.1. Participants

A qualitative study approach was used in which semi-structured interviews were held with fifteen child welfare practitioners (n = 3 men and n = 12 women), including five mobile crisis response team workers (family services), five (child) psychologists/educationalists, three social workers, one family coach, and one child protection worker. As all practitioners had expertise in child welfare, child maltreatment prevention, and assessment procedures, a sufficient degree of data saturation could be assumed [27]. The interviewed practitioners were not involved in any way in the development of ARIJ-Needs, and had no conflicts of interest.

### 2.2. Procedure

A purposive sampling method, specifically expert sampling was used [28]. Practitioners were recruited by contacting the organizations that formed a consortium that was set up to conduct a large-scale child abuse prevention and treatment research project including the current study. The practitioners were informed about the aims and procedure of the study before (informed) consent was obtained. Semi-structured interviews with a mean duration of 37 min were conducted on site at the office of the practitioner, or online through a video call. Practitioners were asked for permission to record the interview and informed participants that all personal data was anonymized for this study.

### 2.3. Instruments

#### 2.3.1. Interviews

The interviews were semi-structured and started with questions about the current decision-making process of practitioners: (1) How are treatment needs assessed in daily practice? (2) How does the decision-making process on appropriate treatment programs takes place? (3) Are there any difficulties in providing appropriate care or interventions for families? Next, a written vignette of a fictitious child protection case that described a variety of family problems was handed out to each participant. This vignette was developed and used in a previous study [14], and is available upon request. The interviewees were asked to identify any care need in the vignette, and to specify what treatment would be appropriate to address the needs they identified. Then, the already developed needs assessment tool (ARIJ-Needs) was presented to each participant and the interviewer made the participants acquainted with the tool. The practitioners were then asked to reperform the needs assessment and to use ARIJ-Needs in this assessment. In the end, several questions were asked on how the practitioners experienced the usability of the tool. The practitioners also evaluated the results of their assessment of needs for which they used ARIJ-Needs.

#### 2.3.2. ARIJ-Needs

ARIJ-Needs is a software application that was designed to support child protection practitioners in the Netherlands in (1) assessing treatment needs and (2) the decision-making process of appropriate care or interventions that target the assessed needs [15]. As described in the Introduction, all dynamic (i.e., changeable) risk factors that are empirically associated with child maltreatment—and can therefore be targeted in interventions—were included in the instrument, and classified into four major categories: parenting factors, family factors, parent factors, and child factors. These risk factor categories guide practitioners in conducting a structured needs assessment, and any factor that a practitioner deems present in a specific case can be selected in the instrument during the assessment. This means that across the four factor categories, practitioners can select multiple factors according to their own clinical judgment. Selecting a risk factor in the instrument means that a factor is deemed to be present in a child protection case. For each need factor that is included in the instrument, a practitioner can request an elaborate description of the factor to facilitate a uniform interpretation of the need factors across practitioners. 

After all identified needs have been selected, interventions that aim to target those needs can be retrieved from the instrument’s database using the “search” button. For every intervention, additional information can be requested in which the aims, the target group, and the level of effectiveness (i.e., based on effectiveness classifications, [29]) of the program are described. In addition, information on the availability of the interventions across regions in the Netherlands can be retrieved. All this information was retrieved from original protocols or manuals of the interventions that were included in the database [15]. The results of the search in terms of the identified needs and the programs targeting those needs can be saved by the practitioner.

### 2.4. Data Analysis

The interviews were transcribed and thereafter analyzed in ATLAS.ti according to the Boeije’s guidelines [30]. In the “open coding stage” of the first few interview codings, different themes were classified into code groups for which a coding scheme was developed (i.e., the “axial coding stage”). Next, new codes were created, or existing codes were integrated with corresponding codes in the “selective coding stage”. The coding process resulted in 159 codes that were classified into 16 “code groups” (i.e., 1: assessing treatment needs in families, 2: deciding on appropriate intervention(s), 3: difficulties in determining what interventions are appropriate, 4: determining additional care for families, 5: applying interventions, 6: evaluation of the intervention(s) suggested by ARIJ-Needs, 7: evaluation of assessing needs using ARIJ-Needs, 8: deciding on the appropriate intervention(s) using ARIJ-Needs, 9: information that is missing in ARIJ-Needs, 10: information that is redundant in ARIJ-Needs, 11: judging the effectiveness of interventions that are part of ARIJ-Needs, 12: user-friendliness of ARIJ-Needs, 13: suggestions for further development of ARIJ-Needs, 14: usability of ARIJ-Needs for practitioners, 15: recommending the use of ARIJ-Needs in child welfare, 16: other suggestions). The first author coded all interviews and flagged any difficulties in the coding procedure. These difficulties were resolved by discussion with all study authors until full consensus was reached. In the end, all authors checked and approved all interview codings. In addition, an independent samples *t*-test was performed to examine differences in the number of assessed need factors between the sampled practitioners that used the ARIJ-Needs and the practitioners that did not use the ARIJ-Needs. 

## 3. Results

### 3.1. Assessing Needs: Current Daily Practice in Child Protection Services

Most practitioners (*n* = 12) usually assess treatment needs at intake with their clients based on their own clinical judgment, after which they often determine a treatment plan based on their own expertise (*n* = 5) (“Structured protocols for determining treatment targets are available, but in reality, all of my colleagues use their own ways.” … “I usually select an intervention that I am familiar with”). Six practitioners mentioned that they usually consult their colleagues for prioritizing needs, and nine practitioners consult their colleagues for determining a treatment plan (“I usually present my case during a weekly meeting after which treatment suggestions are discussed”). Four practitioners mentioned that they consult external authorities to determine an appropriate treatment plan for their assigned case. Three practitioners mentioned that they take specific characteristics of clients (e.g., cognitive abilities or cultural identities) into account in choosing appropriate treatment programs.

### 3.2. Difficulties in Providing Appropriate Care

Most practitioners (*n* = 12) pointed out the long waiting lists for treatment programs in family services as the main barrier to appropriate care for their clients (“The average waiting time for appropriate treatment programs for my clients takes up to several months”). Therefore, out of necessity, clients are offered alternative and less fitting treatment programs. Six practitioners experienced difficulties in working with other care providers and external institutions (“The speed of following-up on a case really depends on the capabilities and willingness of the person that the case is assigned to”). Seven practitioners experienced funding difficulties in providing appropriate care for their clients (“Unnecessary bureaucracy leads to longer waiting periods, which can be frustrating given that you want to be responsive towards your clients”). Three practitioners mentioned that extensive and complicated inclusion criteria of treatment programs can be a barrier to providing the care they think would be most beneficial for their clients. Three practitioners admitted that they are not entirely aware of the wide range of available treatment programs (“I usually select an intervention based on my previous experiences with other clients, but sometimes I wonder, maybe there are other programs that would be more appropriate based on my client’s needs”).

### 3.3. Needs Assessment with and without Using ARIJ-Needs

On average, practitioners assessed significantly more need factors when they used ARIJ-Needs (*M* = 21.67, *SD* = 5.98) than when they based the assessment on their (unstructured) clinical judgment (*M* = 9.07, *SD* = 3.88) (*t* (28) = −6.84, *p* < 0.001). More specifically, practitioners assessed significantly more “family” (e.g., domestic violence, a problematic relationship between care providers, and financial difficulties), “parenting” (e.g., inconsistent parenting, disturbed parent–child interaction patterns, and difficulties in setting up rules and boundaries), and “parent” (e.g., parental stress and aggression regulation issues) need factors (see Table 1).

### 3.4. Evaluation of the Clinical Value of ARIJ-Needs

All practitioners evaluated ARIJ-Needs as helpful in determining family (care) needs. Specifically, practitioners mentioned that the classification of need categories in ARIJ-Needs helpfully structured the needs assessment, which may also provide practical guidance in writing case reports. Practitioners also mentioned that ARIJ-Needs can easily be used within teams of practitioners for discussing individual cases, and that the tool is useable for assessing needs in multi-problem families. However, an overlap in need factors was noted (“For example, in the category “family factors”, I think that some factors in this category can be assigned to the broadly interpretable factor “parenting instability”). Two other practitioners mentioned that the dichotomous answer scale (i.e., yes/no in reporting the presence of a need factor) could be changed into a broader scale to indicate the severity of the client’s needs. Three other practitioners suggested that positive and protective factors (e.g., social support) may also be included in the tool. In addition, other factors suggested for inclusion were the following: information on previous treatment trajectories, complicated divorce of parents, traumatic experiences of children, the appropriate age range of interventions, and the degree to which parents have mentalization skills.

Five practitioners mentioned that the decision-making component (i.e., matching the assessed needs to interventions that target those needs) provided important insights into the many available interventions in child welfare (“It provides a practical overview of the many possibilities in child welfare treatment programs, some of which I had not thought about, or had never heard of”). Three practitioners emphasized the value of providing information on the local availability of interventions which is shown in the results of the instrument. 

All practitioners would like to use ARIJ-Needs in their daily practice (“Without ARIJ-Needs, I am more likely to select an intervention that I have applied before, instead of programs that are lesser known, but potentially more appropriate”). Overall, twelve practitioners evaluated ARIJ-Needs as a user-friendly and comprehensive needs assessment instrument (“The instrument is very practical, especially the classification of need factors in different categories is helpful in the process of assessing treatment needs”). Suggestions for improvement of the instrument were the following: developing a less “basic” and more attractive interface design, developing a web version of the instrument, using less jargon.

## 4. Discussion

ARIJ-Needs is an assessment instrument that supports child protection practitioners in assessing specific treatment needs and in selecting appropriate treatment programs that target those needs. This study is the first to assess the clinical value and utility of a needs assessment instrument that was developed in the Netherlands (ARIJ-Needs) by interviewing child protection practitioners. The results reveal that the decision-making process on selecting treatment programs for families in (Dutch) child protection services is still based on unstructured, clinical judgment, despite the well-known advantages of structured decision-making [12,31]. The practitioners that were interviewed in this study usually selected interventions for their clients based on prior experiences with and referrals to known interventions, or on advice from colleagues. However, (heavily) relying on this form of intuitive thinking may lead to various biases such as the tendency to go for the familiar, the vivid, the “obvious”, and to overlook the unfamiliar, the complex, and the less predictable case information and interventions [32,33].

The results support the idea that structured decision-making facilitates a more holistic approach to treatment settings in which the child’s family and environment are taken into account, which improves practitioners’ analysis of complex situations [34]. Relative to unstructured clinical judgment, practitioners identified significantly more treatment needs in a vignette when they used a needs assessment instrument (ARIJ-Needs). Moreover, practitioners assessed more parent needs (such as delinquent behavior) as well as parenting needs (such a disturbed parent–child relation) when they used the needs assessment tool compared to their unstructured needs assessment. As these needs are more predictive of child maltreatment than child-related factors (see, for instance, [21,35,36]), the current results indicate that ARIJ-Needs supports practitioners in identifying the care needs that are most relevant to address in treatment for families at risk for child maltreatment.

Regarding the usability of ARIJ-Needs, practitioners mentioned that the overview of treatment needs in different categories helpfully structured the needs assessment. In addition, it was mentioned that such an overview can be helpful in efficiently writing CPS reports, which often is an elaborate administrative task that many Dutch child welfare practitioners experience as a burden [37]. According to the practitioners, the decision-making component of ARIJ-Needs (i.e., matching the assessed needs to interventions targeting those needs) provided important insights into the growing range of available interventions in child welfare that many Dutch child welfare practitioners may not be familiar with.

### 4.1. Suggestions for Improving ARIJ-Needs

The interviewed practitioners offered several suggestions for increasing the usability of ARIJ-Needs in clinical practice. First, not all interventions included in the database of ARIJ-Needs are available in all regions of the Netherlands due to differences in care provision policies across Dutch cities and local governments. On the one hand, this might be a barrier to the general usability of ARIJ-Needs. On the other hand, this means that by using the instrument, critical gaps in regional access to treatment programs can be identified by child welfare practitioners. In turn, practitioners can provide consultation to child welfare departments of local governments on which interventions are needed and should become available to meet the needs of their clients. 

Second, practitioners noted that the needs-assessment component in ARIJ-Needs only comprises changeable risk factors, but they would also like to assess protective factors next to risk factors. However, according to the RNR needs principle [11], treatment is most effective when it addresses the dynamic risk factors assessed as being present in a family (i.e., the “needs” or “care needs”) given the theoretical assumption that successfully addressing these factors contributes to a family’s reduction in risk of child maltreatment. Furthermore, research shows that strengthening protective factors may be less effective in preventing recurring child maltreatment in specifically high-risk families [38]. That is, “resilience” as a global construct appears to be rare at the highest levels of risk, and may benefit from a narrower conceptualization focusing on specific outcomes at specific timepoints in development [39]. 

Third, only a limited number of interventions target multidimensional family problems with many and diverse needs, as was the case with the vignette that was presented to participants in this study. The lower the number of risk factors that are selected in the decision-making component of ARIJ-Needs, the more appropriate interventions targeting the selected needs are presented on screen by the decision-making component of the tool. Practitioners mentioned that the dichotomous answer scale could be changed into a broader scale to indicate the severity of the client’s needs. This seems an essential suggestion for improvement, as implementing such a scale would enable practitioners to prioritize the needs that should be targeted with urgency. 

Finally, future research should be undertaken to examine the psychometric quality of ARIJ-Needs. Previous studies have already showed that through applying valid and reliable safety and risk assessment instruments, the decision-making processes in CPS can be strengthened [40,41]. However, the question whether or not the implementation of ARIJ-Needs truly enhances the effectiveness of the decision-making process in child protection and, in turn, improves child maltreatment prevention efforts is important to answer in future research.

### 4.2. Clinical Implications

The barriers to appropriate treatment trajectories that the interviewed practitioners posed (e.g., long waiting lists and insufficient cooperation between institutions) correspond to those found in previous studies on CPS practices in the Netherlands [42,43]. A structured, consistent, and transparent approach to assessing needs using a needs assessment instrument like ARIJ-Needs is likely to contribute to the overall consistency of decision-making in families. This is of particular importance when a family’s involvement with child protection is over a long period of time in which a child and family have contact with numerous practitioners and care providers [44]. It is important to stress here that ARIJ-Needs was not designed with the purpose of replacing the clinical judgment of practitioners, as unique factors that can be case- and/or time-specific—such as urgency and severity—are essential to take into account in deciding on the most appropriate and suitable intervention approach in families that come into contact with CPS [15]. Besides these implications that directly result from this study, it is important to stress that parents and children should be actively involved in decision-making processes in CPS to successfully build a trustful relationship and a positive working alliance [13,32,45].

Further, the first step in the diagnostic process comprises assessing the child’s immediate safety to determine whether safety measures should be taken to safeguard a child. The next steps are assessing the risk for future child maltreatment that informs practitioners about which families should be treated and what level of intensity is required, and assessing the dynamic risk factors (i.e., needs) as described in the current study [15]. In addition, after a needs assessment, an assessment of responsivity factors (e.g., problem denial, treatment motivation, and cognitive abilities) is of equal importance to tailor treatment programs to the unique characteristics of clients to enhance treatment effectiveness [46].

## 5. Conclusions

This study showed that a recently developed needs assessment in the Netherlands (ARIJ-Needs) supports practitioners in the decision-making process on appropriate interventions for families involved in child protection services. Practitioners that used ARIJ-Needs in their assessment identified more treatment needs in a CPS case compared to their clinical, unstructured needs assessment. Particularly, more needs related to the parent(s) and parenting were identified with ARIJ-Needs, which is an important finding as specifically these parent- and parenting-related (dynamic) risk factors are most strongly associated with child maltreatment. The decision-making component of ARIJ-Needs—in which the identified needs are matched to interventions targeting those needs—supported practitioners in selecting appropriate care out of the continuously growing number of available interventions in child welfare.

## Figures and Tables

**Table 1 children-09-01702-t001:** Differences in Identified Dynamic Risk Factors Between Needs Assessment With and Without the Needs Assessment Instrument (ARIJ-Needs).

Needs Domain	Without ARIJ-Needs		With ARIJ-Needs		*t*
	*M*	*SD*	*M*	*SD*	
Family	3.53	1.06	5.87	0.83	−6.70 ***
Child	2.53	2.48	3.60	1.50	−1.43
Parenting	1.73	0.88	6.40	3.38	−5.18 ***
Parent	1.27	1.28	5.80	2.11	−7.11 ***
Total	9.07	3.88	21.67	5.98	−6.84 ***

Note. An independent samples *t*-test was performed for each category of dynamic risk factors (needs) to test the difference in mean number of identified needs between the needs assessments without and with ARIJ-Needs. Using the instrument was scored dichotomously (0 = without ARIJ-Needs, 1 = with ARIJ-Needs), meaning that a negative t value indicates a higher mean number of identified needs with the ARIJ-Needs. *** *p* < 0.001.

## Data Availability

Not applicable.

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
