# Peer review of "Personalizing Child Protection: The Clinical Value and Usability of a Needs Assessment Instrument in The Netherlands"

_children, 2022, doi:10.3390/children9111702_

Round 1

Reviewer 1 Report

Dear Authors,

The article entitled ‘Personalized Child Protection: The Clinical Value and Usability of a Needs Assessment Instrument’ was reviewed for Children.

Evaluating a child who is at risk of maltreatment, and deciding the right appropriate interventions is not easy. An instrument for this process would prevent to overlook any individual needs.

The research was well-written in general in terms of hypothesis, methodological accuracies, results and discussion. However, it needs minor revisions. The study is related to child protection system in Netherlands, so ‘Netherlands’ could be added in the Title. Also the information and the results are not enough to understand ARIF-Needs. Sample questions from ARIJ-Needs and 16 code groups from semi-structured interviews could be given. 

Reviewer 2 Report

1- What type of qualitative and quantitative study was used? Please mention the methods

2- In which way do you analyze your study? In the data analysis section, you describe a qualitative approach, but in "Table 1" you show the mean, SD, and t-test results.

3- Which trustworthiness techniques were used?
